# The Roles of ceRNAs-Mediated Autophagy in Cancer Chemoresistance and Metastasis

**DOI:** 10.3390/cancers12102926

**Published:** 2020-10-11

**Authors:** Huilin Zhang, Bingjian Lu

**Affiliations:** 1Department of Surgical Pathology, Women’s Hospital, School of Medicine, Zhejiang University, Hangzhou 310002, Zhejiang Province, China; huilinzhang@zju.edu.cn; 2Department of Surgical Pathology and Center for Uterine Cancer Diagnosis & Therapy Research of Zhejiang Province, Women’s Hospital, School of Medicine, Zhejiang University, Hangzhou 310002, Zhejiang Province, China

**Keywords:** autophagy, ceRNA, lncRNA, circRNAs, miRNA, chemoresistance, metastasis, pre-metastasis niche

## Abstract

**Simple Summary:**

Chemoresistance and metastasis are the main causes of treatment failure in cancers. Autophagy contribute to the survival and metastasis of cancer cells. Competing endogenous RNA (ceRNA), particularly long non-coding RNAs and circular RNA (circRNA), can bridge the interplay between autophagy and chemoresistance or metastasis in cancers via sponging miRNAs. This review aims to discuss on the function of ceRNA-mediated autophagy in the process of metastasis and chemoresistance in cancers. ceRNA network can sequester the targeted miRNA expression to indirectly upregulate the expression of autophagy-related genes, and thereof participate in autophagy-mediated chemoresistance and metastasis. Our clarification of the mechanism of autophagy regulation in metastasis and chemoresistance may greatly improve the efficacy of chemotherapy and survival in cancer patients. The combination of the tissue-specific miRNA delivery and selective autophagy inhibitors, such as hydroxychloroquine, is attractive to treat cancer patients in the future.

**Abstract:**

Chemoresistance and metastasis are the main causes of treatment failure and unfavorable outcome in cancers. There is a pressing need to reveal their mechanisms and to discover novel therapy targets. Autophagy is composed of a cascade of steps controlled by different autophagy-related genes (ATGs). Accumulating evidence suggests that dysregulated autophagy contributes to chemoresistance and metastasis via competing endogenous RNA (ceRNA) networks including lncRNAs and circRNAs. ceRNAs sequester the targeted miRNA expression to indirectly upregulate ATGs expression, and thereof participate in autophagy-mediated chemoresistance and metastasis. Here, we attempt to summarize the roles of ceRNAs in cancer chemoresistance and metastasis through autophagy regulation.

## 1. Introduction

Autophagy is an evolutionarily conserved system to maintain homeostasis in various cells. In general, there are three main types of autophagy: macroautophagy, microautophagy and mitoautophagy [1]. Mitoautophagy can remove damaged mitochondria specifically by autophagosomes for lysosomal degradation while microautophagy, mainly in plants and fungi, may complete the isolation and uptake of cell components by directly enveloping them with the vacuolar/lysosomal membrane. Macroautophagy is the best-characterized process that aims to remove unwanted or damaged organelles and aggregated proteins by lysosome degradation. In the narrow sense, some scientists (including those in this article) have referred to macroautophagy as autophagy.

The dysregulation of autophagy has been reported in many human diseases, such as infection, cardiovascular and neurodegenerative diseases, and cancers [2]. Metastasis and chemoresistance are two major factors that are associated with cancer recurrence and dismal clinical outcomes. Moreover, metastasis and chemoresistance are closely linked in cancers, as metastatic cancer cells have a propensity for chemoresistance [3,4]. The roles of autophagy in the processes of cancer metastasis and chemoresistance are starting to be recognized. Autophagy is critically involved in many aspects of cancer metastasis, including epithelial–mesenchymal transition (EMT), invasion, migration, anoikis resistance, crosstalk between cancer and stromal cells, and immune suppression [5]. Cancer cells may survive chemotherapy through protective autophagy [6,7,8].

Long non-coding RNAs (lncRNAs) and microRNAs (miRNAs) are non-coding RNAs (ncRNAs) with respective lengths of more than 200 nucleotides and about 22 nucleotides. Both participate in various physiological and pathological processes involving tumorigenesis, such as cell proliferation, apoptosis, EMT, invasion, migration, chemoresistance, and stemness [9]. lncRNAs can regulate gene expression through repressing chromatin components [10], mediating epigenetic silencing, interacting with Polycomb complex, and modulating transcription activation [11]. Considering that there is a complex interplay among messenger RNA (mRNA) and ncRNAs [12,13], lncRNAs also function as miRNA sponges by competitively binding with miRNAs, thereby releasing the targeted mRNAs [14,15]. These lncRNAs are called competing endogenous RNAs (ceRNAs). They can decrease the activity or expression of miRNAs [16,17]. Circular RNA (circRNA) is another kind of ncRNA comprising covalently closed single-stranded loops. Recent advances in RNA sequencing and bioinformatics tools have led to the discovery and identification of thousands of circRNAs and validated their important roles in the pathogenesis of cancers [18,19,20,21]. They can mainly function as ceRNAs to modulate gene transcription by interacting with miRNAs or lncRNAs, sponging miRNAs, or RNA-binding proteins, and rarely can be translated into proteins [22,23,24,25]. In addition to lncRNA and circRNA, other ceRNAs have also been identified by bioinformatics analysis and experimental evidence, such as pseudogenes, mRNA including those expressing 3′-untranslated regions, virus non-coding RNAs, and genomic viral RNAs [26]. Pseudogenes, the relicts of parental gene that are unable to encode full-length proteins, can regulate the expression of parental genes via ceRNA networks, and participate in tumorigenesis [27]. Some mRNAs can function as ceRNAs and can be associated with human diseases such as cardiovascular diseases and glioblastoma multiforme [28,29]. Accumulative data have indicated that ceRNAs, particularly lncRNAs and circRNAs, can bridge the interplay between autophagy and chemoresistance or metastasis in cancers [30,31,32,33,34]. In this review, we will focus on the function of ceRNA-mediated autophagy in carcinogenesis, specifically regarding metastasis and chemoresistance.

## 2. The Biological Process of Autophagy

The process of autophagy is composed of several consecutive steps (Figure 1): autophagy initiation (autophagosome formation), autophagosome nucleation regulated by Beclin-1, autophagosome elongation, and autophagosome fusion with lysosome and degradation [5].

Two serine-threonine protein kinases, the mammalian target of rapamycin (mTOR) and the mammalian homologs of yeast ATG1-Unc-51-lilke kinases 1 (ULK1), are critical for autophagosome formation. mTOR can form mTOR complex 1 (mTORC1) with G protein β-subunit-like protein (GβL), Raptor, and Deptor [35,36], whereas ULK1 interacts with the focal adhesion kinase family interacting protein of 200 kDa (FIP200) and ATG13 to assemble the ULK1 complex [37,38]. Activated mTORC1 can disrupt the ULK1 complex by phosphorylating ULK1 and ATG13 [39,40,41]. Upon energy starvation, activated 5′-AMP-activated protein kinase (AMPK) directly phosphorylates ULK1, but inactivates mTORC1 [37,41]. The inactivation of mTORC1 and the intact ULK1 complex subsequently activate the class III phosphatidylinositol 3-kinase (PIK3C3) complex, containing the core components vacuolar protein sorting 15 and 34 (VSP15, VSP34) and Beclin-1 [42]. The PIK3C3 complex then converts phosphatidylinositol (PtdIns) to phosphatidylinositol 3-phosphate (PtdIns3P), which is recognized by WD-repeat protein interacting with phosphoinositides (WIPIS), the key PtdIns3P effector forming the nascent autophagosome [43,44].

Autophagosome elongation is mainly regulated by two ubiquitin-like conjugation systems: the ATG5–ATG12 complex with the help of ATG7, and LC3 with the assistance of ATG3 and ATG7 [45]. The ATG5–ATG12 complex, which interacts with ATG16L1 on the autophagosome membrane, functions as an E3 ubiquitin ligase-like enzyme in the process of autophagosome elongation [45,46]. The proLC3 (unprocessed LC3) can be processed into cytoplasmic soluble LC3-I after the cleavage by ATG4B. Subsequently, LC3-I interacts with the lipid phosphatidylethanolamine (PE), which is named membrane-binding LC3-II. LC3-II recruits p62/SQSTM1 and TBC1D25/OATL1 to provide a base for selective autophagy. Eventually, the fusion of autophagosome and lysosome generates the autophagolysosomes for degradation [45]. 

## 3. ceRNAs-Regulated Autophagy in Cancer

ceRNAs, acting as miRNA sponges, participate in many aspects of autophagy, from initiation to maturation. They can modulate autophagy phagophore initiation by upregulating mTOR, ULK1, ATG14L, Beclin-1, and autophagy phagophore elongation by upregulating ATG12, ATG5, ATG7, ATG4, and ATG3. Some ceRNAs, such as HOTAIRM1 and HAGLROS, can be involved in multiple steps in the process of autophagy by targeting different genes via multiple binding sites with miRNAs (Table 1). The interplay between ceRNAs in autophagy has gained particular attention in various cancers, particularly in hepatocellular carcinoma (HCC), colorectal cancer, and pancreatic cancer (Figure 1). Rarely, lncRNAs and circRNAs can also regulate autophagy through RNA–protein or RNA–RNA interactions, independently of the roles of ceRNA. For example, lncRNA–ATB can regulate autophagy by activating Yes-associated protein (YAP) and interacting with ATG5 directly, thereby leading to HCC progression [47]. RNA-binding protein human antigen R (HuR) can increase autophagy in intestinal epithelium by upregulating ATG16L1 expression via binding with ATG16L1 mRNA. circPABPN1 can downregulate ATG16L1 to inhibit autophagy by ejecting HuR [48]

### 3.1. HOTAIRM1

HOTAIRM1 functions as a ceRNA to regulate autophagy initiation and elongation through ULK1, ATG3, ATG7, and ATG12 in cancers. A recent study has shown that HOTAIRM1 can enhance autophagosome formation to degrade oncoprotein PML–RARA by sponging miR-20a to upregulate ULK1 in acute promyelocytic leukemia (APL)-ascites mouse model and APL cell lines [52]. Analogously, HOTAIRM1 can play a tumor-suppressor role in ovarian cancers. HOTAIRM1 downregulation has been found to be associated with advanced International Federation of Gynecology & Obstetrics (FIGO) stage, while overexpression inhibits cell proliferation in vivo [82]. On the contrary, HOTAIR has been identified to be an oncogene in HCC, colorectal cancer, and chondrosarcoma. The elevated level of HOTAIR expression has been found in HCC tissues relevant to the normal tissues and to correlate with large tumor size. HOTAIR can enhance autophagy in HCC cells by upregulating ATG3 and ATG7 [83]. In colorectal cancer, HOTAIR upregulated ATG12 expression by sponging miR-93. Knockdown of HOTAIR and ATG12, or overexpression of miR-93, suppressed autophagy and restored radiosensitivity in colorectal cancer cells [58]. In chondrosarcoma, elevated expression of HOTAIR predicted advanced tumor stage and poor survival. HOTAIR knockdown impeded chondrosarcoma progression in vitro and in vivo through inhibiting autophagosome formation by downregulating ATG12 [76]. The conflicting roles of HOTAIRM1 in autophagy and carcinogenesis are determined by the different target miRNAs which may have a tissue-specific expression pattern.

### 3.2. HAGLROS

The overexpression of lncRNA HAGLROS has been found in gastric cancer and HCC and correlated with poor clinical outcomes in patients with these cancers [53,54]. In gastric cancer, HAGLROS was upregulated by the transcription factor STAT3. HAGLROS may inhibit autophagy in two conflicting manners. It can competitively decoy miR-100-5p to antagonize its inhibition on mTOR. On the other hand, it can activate the mTORC1 signaling pathway, an important negative signal of autophagy, via direct interaction with mTORC1. More importantly, the inhibition of autophagy by HAGLROS can promote cell proliferation and migration in gastric cancer cells [53]. In HCC, HAGLROS facilitated cell proliferation, inhibited apoptosis, and enhanced autophagy via regulating the miR-5095/ATG12 axis [54]. 

### 3.3. PVT1

The lncRNA human plasmacytoma variant translocation 1 (PVT1) has been identified as an oncogene and regarded as an indicator for poor survival in pancreatic ductal adenocarcinoma [32], HCC [56], and osteosarcoma [84], although the target genes were not identical in these cancers. PVT1 expression levels were positively associated with that of ULK1 in pancreatic ductal adenocarcinoma tissues. It induced cyto-protective autophagy by targeting ULK1 in vitro and in vivo. Further study showed that PVT1 functioned as an miRNA sponge of miR-20a-5p and restored the expression of ULK1 in the progression of pancreatic ductal adenocarcinoma [32]. In HCC, PVT1 enhanced autophagy by regulating ATG3 expression via functioning as the sponge of miR-365 [56], whereas in osteosarcoma, PVT1 promoted cell migration and invasion through decoying miR-485 [84].

### 3.4. PTENP1

The lncRNA PTENP1, a pseudogene of PTEN, was markedly downregulated in HCC specimens and cell lines [49,50]. As a sponge of miR-17, miR-19b, and miR-20a, PTENP1 can indirectly increase the expression of the miRNA targets including ULK1, ATG7, p62, PTEN, and PHLPP (an inhibitor of AKT); therefore, it can promote HCC progression by provoking pro-death autophagy [49]. In addition, PTENP1 overexpression inhibited migration and invasion in HCC cells through decoying miR-193a-3p, which can activate autophagy by targeting PTEN [50,51].

### 3.5. Other ceRNAs

In non-small-cell lung cancer (NSCLC), KCNQ1OT1 silencing inhibited autophagy and proliferation in vitro and tumor growth in murine xenograft models, but miR-204-5p inhibitor abrogated these inhibitory effects. Both KCNQ1OT1 and ATG3, a direct target of miR-204-5p, were upregulated in NSCLC tissues. KCNQ1OT1 induced autophagy and facilitated NSCLC progression by decoying miR-204-5p to upregulate ATG3 expression [85].

Overexpression of the lncRNA IDH1–AS1 was found in pancreatic cancer tissues and cell lines. IDH1–AS1 promoted autophagy through upregulating ATG5. Bioinformatics analysis indicated that miR-216b-5p shared complementary base pairing with both ATG5 and IDH1-AS1. Therefore, IDH1–AS1 acted as a ceRNA for miR-216b-5p to enhance ATG5 expression and therefore to facilitate autophagy in pancreatic cancers [59]. Similar approaches have been applied to identify that LINC00160 improved autophagy activity in HCC by sequestering miR-132, resulting in the upregulation of PI3K3R, ATG5, and LC3I/II [60].

The expression level of lncRNA SLCO4A–AS1 was upregulated in colorectal cancer tissues, and positively correlated with that of partition-defective 3 (PARD3) [55], a downstream effector of mTOR and AMPK in the initiation of autophagy [86]. Knockdown of SLCO4A–AS1 inhibited autophagy and proliferation in vivo and in vitro by sponging miR-508-3p targeting PARD3 [55].

The lncRNA MALAT1 can inhibit autophagy by decoying miR-15b-5p to regulate MAPK1 expression, thereby activating the MAPK1/mTOR signaling [87]. Consistently, MALAT1 abrogated excessive autophagy in cutaneous squamous cell carcinoma [88].

In acute myeloid leukemia (AML) cells, the lncRNA UCA1 induced autophagy, which was associated with cell proliferation. Mechanistically, UCA1 may function as an miRNA sponge of miR-96-5p that can inhibit ATG7 expression by binding to the 3′-UTR of ATG7 [57]. 

## 4. An Overview of Autophagy-Regulated Cancer Metastasis and Chemoresistance

Emerging evidence has supported the roles of autophagy in the processes of cancer metastasis and chemoresistance. Cancer cells may succeed in metastasis through tumor-promoting autophagy [5], and survival against chemotherapy through protective autophagy [6,7,8].

In most cancers, the activation of autophagy promotes chemoresistance and indicates poor survival. Chemotherapeutic agents induce stress, thus activating autophagy as a cellular adaptive response [89,90]. Recently, it has been demonstrated that DNA damage can induce the process of autophagy and upregulate related genes, essentially resulting in chemoresistance [91]. Therefore, the activation of autophagy becomes a novel molecular mechanism for chemoresistance modulated by some lncRNAs in cancer cells, such as GBCDRlnc1 in gallbladder cancer [92] and MALAT1 in gastric cancer [93]. Under some circumstances, autophagy inhibition may restore chemotherapy sensitivity. For instance, autophagy suppression by Atg7 knockdown enhanced chemosensitivity and prolonged overall survival in AML mouse models [89]. Autophagy mediated by OPN/NF-κB signaling is essential for pancreatic cancer stem cells, while pharmacological inhibition of autophagy increased drug sensitivity [94]. These findings suggest a link between autophagy and cancer stem cells—the root cause for chemoresistance and disease relapse. Additionally, autophagy suppression can enhance chemotherapy response in triple-negative breast cancers (TNBCs) [90]. In HCC cells, miR-541 increased their sensitivity to sorafenib by inhibiting autophagy [95]. Moreover, N6-isopentenyladenosine also improved chemotherapy sensitivity by impairing autophagy in melanoma [96].

Cancer cells experience invasion into the circulation, migration into the pre-metastatic niche, and colonization at the new place [97]. A substantial body of evidence has shown that autophagy is crucial for cancer metastasis through regulating many important steps in this process, such as tumor invasion, migration, EMT, the crosstalk between stromal cells and tumor cells, and immune surveillance [5,98,99,100]. The precision function of autophagy in cancer metastasis remains controversial to date, with tumor-promoting roles in most studies and suppressive roles in a few [101]. A high level of ATG5 is required for metastasis in the mouse models of pancreatic ductal adenocarcinoma [102], while elevated LC3B has been indicated to be associated with invasion and metastasis in solid tumors [103]. Ube2v1, a ubiquitin-conjugating E2 enzyme variant, can facilitate metastasis in colorectal cancer by inhibiting autophagy. Importantly, rapamycin and trehalose treatment may reverse Ube2v1-mediated metastasis in mouse models [104]. In ovarian carcinoma, circMUC16-mediated autophagy promotes metastasis by directly binding to ATG13 and enhancing its expression. Moreover, cirMUC16 could also function as a ceRNA for miR-199a-5p and relieve its inhibitory on Beclin-1 and RUNX1 [81]. Nevertheless, in breast cancer, the expression of autophagy-related genes negatively correlates with pre-metastasis signatures, and thereof restricts tumor metastasis [105,106].

## 5. ceRNA-Mediated Autophagy and Chemoresistance

Chemoresistance is one of the main causes for therapeutic failure in cancers. Dysregulated autophagy [8,107,108], lncRNAs [33,109,110,111], and circRNAs [107,112,113,114,115] have been indicated to be associated with drug resistance in cancers. ceRNAs can play important roles in the regulation of autophagy-mediated chemoresistance in cancers.

Chemoresistance correlates with tumor relapse and contributes to poor clinical outcomes in patients with colorectal cancers. The lncRNA SNHG6 induced 5-fluorouracil (5-FU) resistance in colorectal cancers by regulating an autophagy-related pathway in vitro and in vivo. Bioinformatics analysis and dual-luciferase reporter assay implicated that SNHG6 could competitively bind to miR-26a-5p and upregulate the expression of ULK1, the direct target of miR-26a-5p, thereof leading to autophagy activation and 5-FU resistance in colorectal cancer cells [80]. Consistently, SNHG6 upregulated EZH2 expression and induced EMT, migration, and invasion by binding to miR-26a in colorectal cancers [116]. These studies indicate that SNHG6 may serve as a promising therapeutic target for colorectal cancers.

The lncRNA XIST was overexpressed in NSCLC tissues and cisplatin-resistant A549 cell lines. XIST silencing improved chemotherapy sensitivity by decreasing autophagy. XIST enhanced ATG7 expression by sponging miR-17, thereof contributing to autophagy-mediated drug resistance in NSCLC [63]. Moreover, the silencing of XIST impeded metastasis in NSCLC by sponging other miRNAs [117,118]. It can facilitate TGF-β-induced EMT and metastasis via regulating miR-367 and miR-141 that target ZEB2 [118], or contribute to cell proliferation, migration, and invasion as a ceRNA of miR-374a to modulate LARP1 expression [117]. Similar to XIST, upregulated lnc-ROR expression induced EMT, migration, and invasion in pancreatic cancers [119]. lnc-ROR knockdown restored drug sensitivity by increasing basal autophagy in pancreatic cancer cells via sponging miR-124, which regulates the PTBP1/PKM2 axis [61]. It has been clarified that PKM2 can activate autophagy through phosphorylating Beclin-1 [120]. Given that cancer cells experiencing EMT and metastasis are usually resistant to chemotherapy, we may suggest that XIST and lnc-ROR promote ceRNA-mediated autophagy to facilitate EMT and metastasis, and may be associated with chemoresistance in NSCLC. The lncRNA UCA1 was found to be overexpressed in bladder cancers. UCA1 knockdown suppressed cell proliferation, invasion, migration, and chemoresistance by sponging miR-582-5p to attenuate its inhibition of ATG7 expression [62].

The lncRNA bladder cancer-associated transcript 1 (BLACAT1) regulated chemoresistance in NSCLC via the miR-17/ATG7 axis. Silencing of BLACAT1 alleviated drug resistance in vivo. RIP and RNA pull-down assays confirmed the direct interaction between BLACAT1and miR-17. BLACAT1 promoted ATG7 expression through miR-17 to induce autophagy and enhance chemoresistance in NSCLC cells [64]. Similar to BLACAT1, KCNQ1OT1 overexpression was associated with decreased chemotherapy sensitivity and poor prognosis in colon cancers. The study in vitro further indicated that KCNQ1OT1 induced autophagy and chemotherapy resistance via upregulating ATG4B expression by sponging miR-34a [65].

lncRNA MALAT1 expression was increased in chemoresistant gastric cancer cells and colorectal cancer cells, and was positively related with autophagy activity [66,121,122]. Emerging data have indicated that MALAT1 may regulate autophagosome maturation through a ceRNA mechanism, contributing to chemoresistance in gastric and colorectal cancers. MALAT1 induced CDDP resistance by enhancing autophagy in gastric cancer cells, which could be reversed by miR-30b overexpression. MALAT1 sequestered miR-30e and miR-30b to upregulate ATG5 expression—a direct target of both miRNAs [66,67]. In colorectal cancer, MALAT1 promoted autophagy activity by acting as a ceRNA with miR-101 [121]. Of note, miR-101 is a potent autophagy inhibitor which targets ATG4D, a member of the ATG4 family which is essential for LC3 processing [123]. Further work is required to validate the potential relationship between MALAT1 and ATG4D in colorectal cancers.

In addition, apatinib, a small-molecule inhibitor of vascular endothelial growth factor receptor 2, is helpful in treating gastric cancers. However, in the clinical setting, some patients received a reduced dose of apatinib in terms of their intolerability and severe complications [124]. Proper intervention is a pressing need to help patients achieve great benefits from apatinib treatment. circCGAP1 knockdown increased apatinib sensitivity in gastric cancer cells by autophagy inhibition. Mechanistically, circCGAP1 sponged miR-3657 to upregulate ATG7, the direct target of miR-3657, leading to autophagy activation [125]. Therefore, specific blockage of the circRACGAP1–miR-3657–ATG7 axis is critical for the regulation of apatinib sensitivity in gastric cancers.

Increased expression of lncRNA NEAT1 was found in various cancers, such as HCC [70], anaplastic thyroid carcinoma [71], and colorectal cancer cells [69], and was correlated with poor prognosis in these cancers. NEAT promoted autophagy-mediated chemoresistance as a ceRNA targeting the miR-204/ATG3 axis in HCC [70], miR-9-5p/SPAG9 in anaplastic thyroid carcinoma [71], and miR-34a/HMGB1, ATG9A, and ATG4B in colorectal cancers [69]. Notably, SPAG9 is critical for lysosome localization and autophagy [126], while HMGB1 initiates autophagosomes by binding with Beclin-1 [127]. ATG9A plays essential roles in the delivery of lipids or proteins to the initiating sites of autophagy [128]. Interestingly, the NEAT1/miR-29b/ATG9A axis was modulated by insulin-like growth factor binding protein-related protein 1 (IGFBPrP1) in hepatic stellate cell autophagy [129]. However, the relationship between the NEAT1/miR-29b/ATG9A axis and IGFBPrP1 has not been clarified yet. In addition to NEAT1, circEIF6 is another ceRNA that bridges autophagy and chemoresistance in anaplastic thyroid carcinomas. circEIF6 knockdown can re-sensitize anaplastic thyroid carcinoma cells to cisplatin, and its overexpression increased cisplatin-induced autophagy and inhibited cell apoptosis by targeting the miR-144-3p/TGF-α axis [72]. 

circCDYL and circABCB10 are two potential biomarkers for chemoresistance in breast cancers. Increased circCDYL level in both tissue and serum samples from breast cancer patients was associated with unfavorable clinical outcomes and drug resistance. circCDYL-induced proliferation in breast cancers depended on autophagy via sponging miR-1275 to upregulate the expression of ATG7 and ULK1 [73]. In an analysis of 30 pairs of paclitaxel-resistant and sensitive breast cancers, circABCB10 expression was positively correlated with paclitaxel resistance. Consistently, silencing of circABCB10 restored paclitaxel sensitivity by inducing apoptosis and inhibiting invasion and autophagy in chemoresistant breast cancer cells through the let-7a-5a/dual specificity phosphatase 7 (DUSP7) axis [74]. Likewise, ceRNA-mediated autophagy also regulated drug resistance in renal cell carcinomas. circ_0035483 overexpression promoted gemcitabine-induced autophagy and drug resistance in renal clear cell carcinoma by upregulating CCNB1 expression by sponging miR-335 [75].

Taken together, the current data demonstrate that ceRNAs can participate in many steps of autophagy by upregulating some key molecules, such as ULK1, ATG3, ATG7, and ATG4, resulting in chemoresistance in various cancers. Most ceRNAs are expressed highly in cancer tissues and enhance autophagy, resulting in chemoresistance. EMT, metastasis, cancer stem cells, proliferation, and apoptosis are the major mechanisms of chemoresistance [130], nevertheless, no evidence has indicated that the ceRNAs-mediated autophagy plays direct roles in chemoresistance and these related processes (Figure 2).

lncRNA HULC and eosinophil granule ontogeny transcript (EGOT) were found to be associated with chemoresistance through autophagy regulation independently of ceRNA. HULC was markedly downregulated in gastric cancer cells with cisplatin resistance. RNA pull-down and RIP assays identified the interaction between HULC and FoxM1. Silencing of HULC inhibited autophagy and reduced cisplatin resistance through regulating FoxM1 [48]. Intriguingly, HULC indirectly increased USP22 expression via epigenetic or transcriptional modulation of miR-6825-5p, miR-6845-5p, and miR-6886-3p rather than sponging these molecules in HCC cells [131]. USP22 upregulated Sirt1 expression by inhibiting its ubiquitin-mediated degradation. Sirt1 deacetylated ATG5 and ATG7, leading to autophagy activation. EGOT, an antisense intronic lncRNA derived from lncRNA ITPR1, promoted autophagy to sensitize paclitaxel cytotoxicity in breast cancers by upregulating ITPR1 expression through RNA–RNA and RNA–protein interactions [132].

## 6. ceRNA-Mediated Autophagy and Metastasis

The roles of ncRNAs such as lncRNAs [133,134], circRNAs [73,135], and miRNAs [133,136] have been recognized in cancer metastasis and autophagy. Figure 3 summarizes the potential roles of ceRNA-mediated autophagy in modulating metastasis. The microenvironment in the metastatic site is referred to as the pre-metastasis niche, which is prerequisite for cancer cells to colonize. Stromal cells including cancer-associated fibroblasts (CAFs), tumor-associated macrophages (TAMs), and other immunocytes as well as exosomes are indispensable for the formation of the pre-metastasis niche [137,138]. Therefore, ceRNA-mediated autophagy in the microenvironment is potentially crucial for regulating cancer metastasis.

Serval lncRNAs were found to be associated with cancer metastasis and advanced TNM stage, such as HCG11 [78] and MCM3AP-AS [79] in HCC, and SNHG6 [68,139] in osteosarcoma. They functioned as ceRNAs promoting autophagosome maturation by upregulating ATG12 or ATG7. HCG11 knockdown inhibited cell proliferation, metastasis, and autophagy in HCC cells and murine xenografts. Interestingly, miR-26a-5p deletion or ATG12 overexpression could rescue the effects caused by HCG11 silencing. miR-26a-5p can bind to both HCG11 and ATG12. As a collar, the biological functions of HCG11 may be obtained through sponging miR-26a-5p to antagonize its inhibitory effect on ATG12 [78]. MCM3AP-AS1 knockdown inhibited HCC cell invasion and lymphatic vessel formation. Bioinformatics analysis and luciferase reporter assay validated the direct interaction between MCM3AP-AS1 and miR-455. Gain- or loss-of-function studies showed that an miR-455 mimic impaired cell invasion and lymphatic vessel formation, which could be rescued by ATG7 overexpression. Therefore, MCM3AP-AS1 promoted metastasis via regulating the miR-455/ATG7 axis [79]. Silencing of SNHG6 suppressed cell proliferation and invasion, and induced autophagy in osteosarcoma cells, possibly by sponging miR-26a-5p to restore the expression of its target ULK1 [68]. lncRNAs SNHG11 and SNHG15 induced proliferation, apoptosis, migration, and autophagy in HCC cells through sponging miR-184 [140] and miR-141 [141], respectively.

Increased circMUC16 expression has been associated with advanced tumor stage in ovarian carcinomas. circMUC16 knockdown or overexpression showed that circMUC16 promoted autophagy flux to facilitate invasion and metastasis in ovarian carcinomas. Further studies indicated that circMUC16 upregulated Beclin-1, RUNX1 expression via sponging miR-199a-5p, and ATG13 level by direct interaction. RUNX1 facilitated circMUC16 expression in a positive feedback. circMUC16 is a promising prognostic biomarker for metastasis in ovarian carcinomas [81].

CAFs and TAMs are two major components in the microenvironment in the metastatic site. Regarding the crosstalk between autophagy among cancer cells, CAFs and TAMs through a ceRNA mechanism have emerged to be recognized in cancer metastasis. HOTAIR sponged miR-454-3p and indirectly upregulated ATG12 to promote cancer progression in chondrosarcoma cells [76]. In breast cancer cells, CAFs induced metastasis by enhancing the expression of HOTAIR transcriptionally, of which the promoter was bound with SMAD2/3/4. Silencing of HOTAIR impeded breast cancer cell growth and lung metastasis in vivo [142]. CAFs upregulated UCA1 expression in colorectal cancer cells, possibly through mTOR-related pathways [143], which was crucial for the pre-metastatic niche. UCA1 overexpression, induced by TAM-derived CCL18 (C-C motif chemokine ligand 18) [144], was closely associated with pulmonary metastasis and unfavorable clinical outcomes in osteosarcomas [145]. The lncRNA MALAT1 alleviated LAMP1 inhibition and promoted autophagy in macrophages by competitively binding to miR-23-3p [146]. ceRNA-mediated autophagy may be vital for TAM in immune regulation in the metastatic niche, despite the absence of direct evidence at present.

Exosomes, extracellular vesicles containing ncRNAs and proteins, are crucial for cellular communication in tumorigenesis in autocrine, paracrine, and endocrine manners [147]. lncRNAs such as PVT1 and UCA1 can be packaged into exosomes to link the crosstalk between stromal cells and tumor cells partly through autophagy regulation, thereby contributing to pre-metastasis niche formation [148,149]. PVT1 promoted tumor cell metastasis by upregulating autophagy. Moreover, PVT1 was encapsulated into bone marrow mesenchymal stem cells (BMSCs)-derived exosomes and promoted osteosarcoma cell proliferation and migration [148]. The elevated UCA1 upregulated ATG7 in bladder cancers by sponging miR-582-5p [62]. UCA1 expression increased in hypoxic exosomes from bladder cells compared to non-hypoxic exosomes and serum-derived exosomes from bladder cancer patients [150]. The hypoxic exosomes facilitated bladder cancer progression through EMT both in vivo and in vitro [62], which may be caused by UCA1/miR-582-5p/ATG7-mediated autophagy. In pancreatic cancer, lnc-ROR modulated autophagy-related chemoresistance through a ceRNA mechanism and induced EMT by upregulating ZEB1, resulting in cell migration and invasion [61,119]. Interestingly, lnc-ROR was found in the thyroid cancer-stem-like-cell-derived exosomes that regulated EMT and remodeled the tumor microenvironment and pre-metastasis niche [149]. Given that the inhibition of autophagy suppressed cancer metastasis and EMT [151], we postulated that lnc-ROR might regulate EMT and metastasis through a ceRNA-mediated autophagy in cancers.

Hypoxia is a key microenvironmental factor in the pre-metastasis niche formation [152,153]. Hypoxia can promote autophagy through a ceRNA mechanism. lncRNA PVT1 facilitated hypoxia-induced autophagy via PVT1/miR-152-ATG14 pathway in hepatic stellate cells (liver-specific mesenchymal cells that can be transformed to fibroblasts) [154]. Importantly, PVT1-induced autophagy has been involved in the proliferation, migration, and angiogenesis of gliomas. Bioinformatics analysis and dual-luciferase assay revealed that PVT1 restored the expression of ATG7 and Beclin-1 via a ceRNA mechanism targeting miR-186 [77]. However, no direct evidence is available for hypoxia-induced autophagy in cancer metastasis to date.

## 7. ceRNAs as Potential Therapeutic Targets in Cancers

As shown in Table 1, most of the ceRNAs involving autophagy are oncogenes and are expressed highly in cancer tissues compared to normal adjacent tissues—especially in HCC, colorectal cancer, and pancreatic cancer (Table 2), which are among the most devastating cancers worldwide [155,156,157]. Upregulation of HAGLROS [54], PVT1 [56], LINC00160 [60], NEAT1 [70], HCG11 [78], and MCM3AP-AS [79] in cancers were associated with poor clinical outcomes in patients with HCC. The oncogenic roles of LINC00160 and HCG11 have been confirmed not only in vitro but also in murine xenograft models. Likewise, SLCO4A-AS1 [55] and HOTAIR [58] were found to be overexpressed in colorectal cancers, and acted as oncogenes in in vitro and in vivo studies. PVT1 overexpression indicated poor clinical outcomes in patients with pancreatic cancers [32]. Its oncogenic roles were confirmed through in vitro and in vivo experiments, and its interaction with miRNAs was confirmed by the RIP assay. 

These ceRNAs are suggested to induce protective autophagy and to promote cancer progression, chemoresistance, and metastasis. Therefore, they and their related pathways have a great potential to serve as therapeutic targets. It has been reported that miRNAs could be delivered through monomethoxy-poly-poly nanoparticles [158]. We propose that the miRNAs sponged by those ceRNAs might be delivered to the cancers to correct the interaction among miRNA, ceRNA, and target genes, thereof alleviating the protective autophagy and eventually achieving therapeutic effects such as the restoration of sensitivity to chemotherapy and inhibition of metastasis. Moreover, CRISPR-based gene-editing tools are potential applicable treatment strategies [159]. The CRISPR/Cas9 system may be designed to downregulate the ceRNAs in cancers. One major limitation of these systems is adverse reactions owing to a lack of tissue specificity. The use of tissue-specific promoters may overcome these problems. Chloroquine and hydroxychloroquine were observed to have anti-tumor effects by targeting autophagy [160]. Moreover, multiple groups of data from Phase I/II clinical trials have confirmed that hydroxychloroquine selectively targets autophagy in cancer patients [161,162]. It would be attractive to combine the tissue-specific miRNA delivery or CRISPR/Cas9 system with hydroxychloroquine, the selective autophagy inhibitor, to treat cancer patients in the future.

## 8. Conclusion and Future Perspectives

In conclusion, ceRNAs play important roles in many human cancers. Most ceRNAs positively regulate autophagy unless they sponge miRNAs targeting mTOR. Nearly all ceRNAs function as oncogenes, with rare exceptions (Table 1).

Most metastatic cancer cells are chemoresistant [3,163]. It is prerequisite to explore the underlying molecular link between metastasis and chemotherapy. Many factors significantly contribute to drug resistance, such as cancer cell heterogeneity, drug efflux, strength of DNA repair, cancer cell stemness, and dysregulated proliferation/apoptosis pathways [164,165,166]. Under unfavorable circumstances, autophagy activation can protect cells from stress-induced injury [167,168]. Unequivocally, chemotherapy drugs can activate autophagy in cancer cells [169]. The metastasis of cancer harbors some key events, such as EMT, invasion into vessels, resistance to anoikis in circulation, and travelling to the secondary site [99]. During this process, cancer cells struggle to survive against various stresses. Likewise, autophagy also plays an important part in metastasis against these stresses [170]. Moreover, even in the secondary site, it is important to build up an appropriate microenvironment—that is, the pre-metastasis niche [171,172]. Autophagy is essential for niche formation owing to its biological effects on stromal cells, immune escape, angiogenesis, and lymphangiogenesis [171]. Therefore, further clarification of the mechanism of autophagy regulation in metastasis and chemoresistance may greatly improve the efficacy of chemotherapy and survival in cancer patients.

lncRNAs, circRNAs, and miRNAs are critical in tumorigenesis and may serve as promising therapeutic targets and biomarkers for cancers [9,173,174]. ceRNAs communicate with different types of RNAs in cancer progression, forming the network among lncRNAs, circRNAs, miRNAs, and mRNAs [15,131]. In this manuscript, we summarized the role of ceRNA-regulated autophagy in chemoresistance and metastasis. ceRNAs can regulate the expression of autophagy-related genes via binding with some miRNAs. These genes include mTOR, ULK1, ATG3, ATG4, ATG5, ATG7, ATG12, ATG13, and ATG14, which are critical for autophagosome initiation, elongation, and fusion with lysosomes (Figure 1). 

To date, ceRNA-mediated autophagy is mostly known as a positive regulator of chemoresistance and metastasis. These ceRNAs can promote chemoresistance through proliferation inhibition and apoptosis induction, and facilitate metastasis by inducing EMT, invasion, migration, angiogenesis, and modulating the pre-metastasis niche (Table 1; Figure 2 and Figure 3). Some ceRNAs can be packaged in exosomes which are derived from cancer cells or bone marrow mesenchymal stem cells. The exosomal ceRNAs contribute to optimizing the microenvironment for cancer metastasis. However, more work is required to uncover how ceRNA-mediated autophagy regulates chemoresistance and metastasis in cancers. Particular issues might be focused on the potential role of ceRNAs in the crosstalk among cancer cells and the tumor microenvironment composed of stromal cells, endothelial cells (angiogenesis), and immunocytes. Nevertheless, several critical issues should first be addressed.

The ceRNA theory is supported by a great body of experimental data on lncRNAs and circRNAs. The studies are invariably associated with ectopic vectors which produce much higher than physiological levels in cancer cells, and unavoidably present non-physiological phenotypes [175]. The stoichiometry required for a ceRNA, a competitor for miRNAs, includes a substantially higher endogenous content and affinity with linear transcripts over miRNAs; therefore, it is very unlikely that a low-level ceRNA or physiological changes in the expression of an individual ceRNA would effectively regulate a far more abundant miRNA target. In fact, the abundance of ceRNAs such as lncRNA and circRNA in cancers has always been a major concern. The absolute levels of functional ceRNAs in cancer tissues have not been well addressed to date, owing to the difficulty in accurately quantifying ceRNAs and due to some technical problems in tissue preparation. The unification of accurate quantification becomes a prerequisite for ceRNA studies in cancers before we can find a functional ceRNA with considerably high contents.

A ceRNA may target multiple miRNAs, and the potential target miRNAs may also have several target genes (Table 1). It is reasonable to reckon that a subtle alteration of the miRNA by its upstream ceRNA may have a significant effect on the target genes in the process of autophagy. The development of a ceRNA network is crucial for better understanding the molecular mechanisms and potential utilities underlying gene expression regulated by ceRNAs. However, ceRNA networks have so far predominantly been constructed by bioinformatic prediction, with very limited experimental data. Another important issue with regard to the interaction between ceRNA and miRNA is the problem of “degradation” or “inhibition”, which is crucial for ceRNA function [175]. There is still conflicting evidence describing the effect of ceRNA–protein complexes on the degradation of miRNA after binding. It may be determined by the status of the binding sites between circRNAs and miRNAs: “degradation” in the case of completely complementary, or “inhibition” when partially matched [176].

Bioinformatic analysis, dual-luciferase assay, and methods measuring gene expression (e.g., reverse transcription-PCR (qRT-PCR) and Western blot) are the most commonly applied approaches in ceRNA investigations [26]. Results from bioinformatic prediction should be validated by experimental data. Dual-luciferase assay and gene expression analysis are not direct and sound methods to establish the interaction among ceRNA, miRNA, and mRNA. The reliable methods are currently RIP and RNA pull-down. However, as listed in Table 2, both methods have only been applied in a small number of studies. Importantly, the experimental results are obtained from gain- or loss-of-function assays in vitro and, uncommonly, in vivo. The technical limitations and defects should be appreciated in the assessment of the current studies. From the perspective of clinical relevance, future elaborate work in primary-cancer-derived organoids which repeat the pathological features of cancer [177], and studies in vivo will consolidate the interaction among ceRNA, miRNA, and RNA; therefore, they will ultimately provide great insights into carcinogenesis including autophagy-related cancer chemoresistance and metastasis.

## Figures and Tables

**Figure 1 cancers-12-02926-f001:**
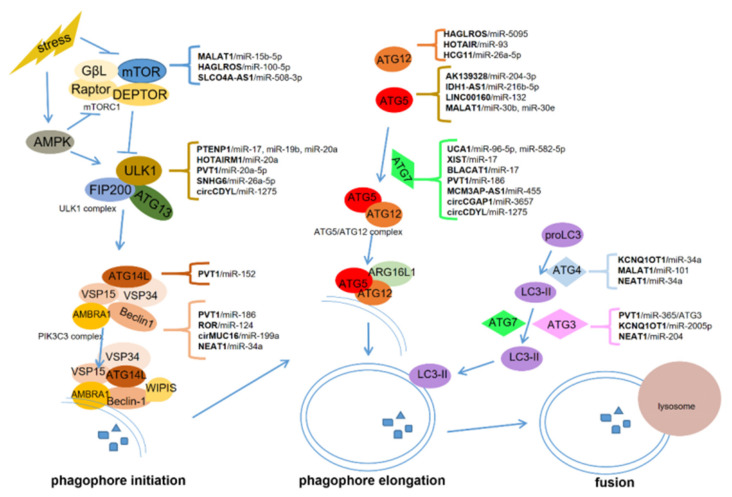
Competing endogenous RNAs (ceRNAs) regulate autophagy in cancers. ceRNAs regulate autophagosome initiation or phagophore elongation by upregulating mTOR, ULK1, ATG13, ATG14L and Beclin-1, or ATG12, ATG5, ATG7, ATG4, and ATG3, respectively.

**Figure 2 cancers-12-02926-f002:**
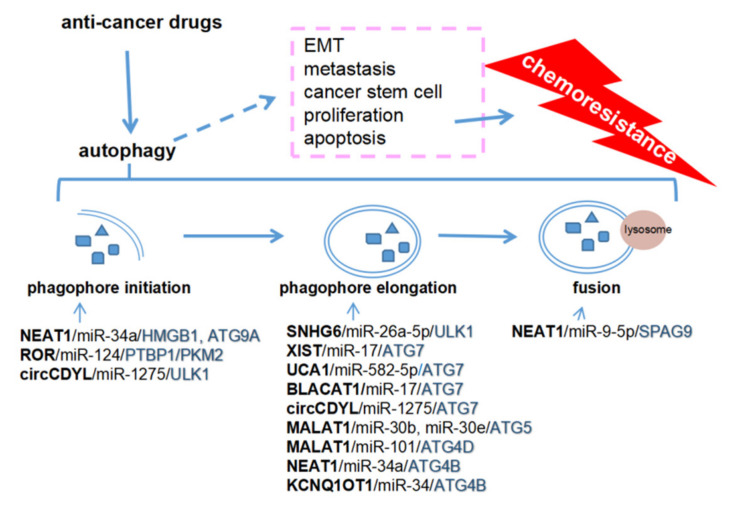
Competing endogenous RNAs (ceRNAs) regulate autophagy to promote chemoresistance. NEAT1 and ROR are involved in the initiation of phagophore formation, and SNHG6, XIST, UCA1, BLACAT1, KCNQ1OT1, MALAT1, and NEAT1 in the elongation of autophagosome. NEAT1 also regulates lysosome localization by upregulating SPAG9. ceRNAs-mediated autophagy may promote chemoresistance via EMT, metastasis, cancer stem cells, proliferation, and apoptosis.

**Figure 3 cancers-12-02926-f003:**
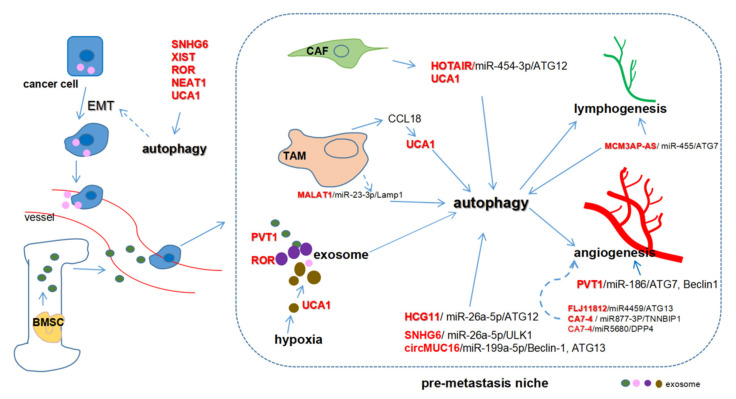
Competing endogenous RNAs (ceRNAs) regulate metastasis through autophagy. Cancer-associated fibroblasts (CAFs) increase HOTAIR and UCA1 expression while tumor-associated macrophages (TAMs) upregulate UCA1 through CCL18. MALAT1/miR-23-3p regulates autophagy in macrophages. MCM3AP-ASinduces lymphatic vessel formation while PVT1 promotes angiogenesis by sponging miR-186. FLJ11812- and CA7-4-mediated autophagy may be essential for angiogenesis. PVT1, ROR, and UCA1 can be packaged in exosomes to promote metastasis. PVT1 can be encapsulated in exosomes derived from bone marrow mesenchymal stem cells (BMSC) while exosomal UCA1 is induced substantially by hypoxia.

**Table 1 cancers-12-02926-t001:** ceRNAs regulate autophagy in cancers.

Categories	ceRNA	miRNA	Target Genes	Cancer Types	Functions	References
Tumor suppressor	PTENP1	miR-17, miR-19b, miR-20a	ULK1, ATG7, p62, PTEN, and PHLPP	HCC	Inhibit proliferation, invasion, and metastasis	[49]
	PTENP1	miR-193a-3p	PTEN	HCC	Inhibit invasion and migration	[50,51]
	HOTAIRM1	miR-20a	ULK1	Acute promyelocytic leukemia	Inhibit proliferation and invasion	[52]
Oncogene	HAGLROS	miR-100-5p	mTOR	Gastric cancer	Promote proliferation, invasion, and migration	[53]
	HAGLROS	miR-5095	ATG12	HCC	Promote proliferation and inhibit apoptosis	[54]
	SLCO4A–AS1	miR-508-3p	PARD3	Colorectal cancer	Promote proliferation	[55]
	PVT1	miR-20a-5p	ULK1	Pancreatic ductal adenocarcinoma	Promote growth	[32]
	PVT1	miR-365	ATG3	HCC	Promote proliferation	[56]
	UCA1	miR-96-5p	ATG7	Acute myeloid leukemia	Promote proliferation	[57]
	HOTAIR	miR-93	ATG12	Colorectal cancer	Promote radioresistance	[58]
	IDH1–AS1	miR-216b-5p	ATG5	Pancreatic cancer	Promote growth	[59]
	LINC00160	miR-132	PIK3R/ATG5	HCC	Promote chemoresistance	[60]
	lnc-ROR	miR-124	PTBP1/PKM2/Beclin-1	Pancreatic cancer	Promote chemoresistance, EMT, metastasis	[61]
	UCA1	miR-582-5p	ATG7	Bladder cancer	Promote chemoresistance, proliferation, metastasis	[62]
	XIST	miR-17	ATG7	NSCLC	Promote chemoresistance, proliferation, EMT, metastasis	[63]
	BLACAT1	miR-17	ATG7	NSCLC	Promote chemoresistance	[64]
	KCNQ1OT1	miR-34a	ATG4B	Colon cancer	promote chemoresistance	[65]
	MALAT1	miR-30b, miR-30e	ATG5	Gastric cancer	Promote chemoresistance	[66,67]
	SNHG6	miR-26a-5p	ULK1	Osteosarcoma	Promote chemoresistance, EMT, metastasis	[68]
	NEAT1	miR-34a	HMGB1, ATG9A, and ATG4B	Colorectal cancer	Promote chemoresistance	[69]
	NEAT1	miR-204	ATG3	HCC	Promote chemoresistance	[70]
	NEAT1	miR-9-5p	SPAG9	Anaplastic thyroid carcinoma	Promote chemoresistance	[71]
	circEIF6	miR-144-3p	TGF-α	Anaplastic thyroid carcinoma	Promote chemoresistance	[72]
	circCDYL	miR-1275	ATG7, ULK1	Breast cancer	Promote chemoresistance	[73]
	circABCB10	let-7a-5a	DUSP7	Breast cancer	Promote chemoresistance	[74]
	circ_0035483	miR-335	CCNB1	Renal clear cell carcinoma	Promote chemoresistance	[75]
	HOTAIR	miR-454-3p	ATG12	Chondrosarcomal cancer	Promote proliferation, metastasis	[76]
	PVT1	miR-186	ATG7, Beclin-1	Glioma	Promote proliferation, metastasis, angiogenesis	[77]
	HCG11	miR-26a-5p	ATG12	HCC	Promote proliferation, metastasis	[78]
	MCM3AP–AS	miR-455	ATG7	HCC	Promote lymphogenesis, metastasis	[79]
	SNHG6	miR-26a-5p	ULK1	Osteosarcoma	Promote proliferation, metastasis	[80]
	circMUC16	miR-199a-5p	ATG13	Epithelial ovarian cancer	Promote metastasis	[81]

Abbreviations: HCC: hepatocellular carcinoma; NSCLC: non-small-cell lung cancer; EMT: epithelial–mesenchymal transition.

**Table 2 cancers-12-02926-t002:** ceRNAs regulate autophagy in HCC, colorectal cancer, and pancreatic cancer.

Cancer Types	ceRNA	miRNAs	Target Genes	Number of Clinical Samples	In Vivo Studies	RIP/RNA Pull-Down	Reference
HCC	HAGLROS	miR-5095	ATG12	68	N	Y	[54]
	PVT1	miR-365	ATG3	80	N	N	[56]
	LINC00160	miR-132	PIK3R/ATG5	68	Y	Y	[60]
	NEAT1	miR-204	ATG3	(TCGA)	N	Y	[70]
	HCG11	miR-26a-5p	ATG12	65	Y	N	[78]
	MCM3AP-AS	miR-455	ATG7	25	N	N	[79]
Colorectal cancer	SLCO4A-AS1	miR-508-3p	PARD3	23	Y	Y	[55]
	HOTAIR	miR-93	ATG12	71	Y	N	[58]
	NEAT1	miR-34a	HMGB1, ATG9A, and ATG4B	55	N	N	[69]
Pancreatic cancer	PVT1	miR-20a-5p	ULK1	68	Y	Y	[32]
	lnc-ROR	miR-124	PTBP1/PKM2/Beclin-1	31	N	N	[61]

Abbreviations: HCC: hepatocellular carcinoma; N: not done; Y: yes.

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
