# Peer review of "The Roles of ceRNAs-Mediated Autophagy in Cancer Chemoresistance and Metastasis"

_cancers, 2020, doi:10.3390/cancers12102926_

Round 1

Reviewer 1 Report

The study by Zhang and Lu provides a very updated review on the role of competing endogenous RNAs and other ncRNAs in autophagy and how this regulation can affect cancer. The review is well written and certainly of interest for both basic and clinical researchers. Several points may be revised by the authors in order to provide mor clarity to the manuscript

  1. Please revised typo errors (g autophagy in Figure 2, etc…) and grammar.
  2. Authors should revised the role of HAGLROS. The authors state based on ref#45 that “It modulated autophagy in two conflicting manners. In one hand, HAGLROS promoted autophagy by competitively decoying miR-100-5p to antagonize its inhibition on mTOR.” One can understand that it may increase or inhibit autophagy, but in fact I understand from this work that this lncRNA inhibits autophagy acting as a ceRNA or directly increasing the activity of mTORC1 by binding to different proteins of the complex. Please review this part.
  3. This review offers a large amount of information that has to be better defined and organized. The authors have to create well defined subsections (with a title) and present the results whether by diseases or by lncRNAs/ceRNAs (probably the best option as authors did in the 4th section). This will add more clarity to the review. For example, in the second section (“CeRNAs regulated autophagy in cancer”) sometimes paragraphs are dedicated to lncRNAs (p.3, last paragraph), sometimes to groups of proteins (p. 4 second paragraph), sometimes to the disease (p.4 3rd paragraph). In addition, in p.4 last paragraph, authors give an overview of lncRNAs that do not act as ceRNAs. Perhaps this last paragraph could be extended to PVT1 for example that besides its role as a ceRNA also has a direct effect on ULK1 (p.4 first paragraph). The same occurs in section 5, where paragraphs describing lncRNAs in autophagy and metastasis are mixed with others on the role of hypoxia or lncRNAs contained in exosomes. Perhaps these two paragraphs could be placed at the end of the section after describing the different lncRNAs affecting autophagy and metastasis.
  4. An important role of reviews besides giving an updated vision of the subject is to also give a critical view on the literature together with the most prominent perspectives. The first objective is reached with many very recent data (2020) and this is certainly a very positive point in this review, however I miss a more personal involvement of the authors concerning the discussion of the points addressed in this review.
  5. I would add a specific section describing the potential use of ceRNAs or miRNAs as therapeutic targets or tools for autophagy in the context of cancer.

Author Response

Referee: 1

Comments to the Author

  1. Please revised typo errors (g autophagy in Figure 2, etc…) and grammar

Reply: The typo errors have been corrected. 

  1. Authors should revised the role of HAGLROS. The authors state based on ref#45 that “It modulated autophagy in two conflicting manners. In one hand, HAGLROS promoted autophagy by competitively decoying miR-100-5p to antagonize its inhibition on mTOR.” One can understand that it may increase or inhibit autophagy, but in fact I understand from this work that this lncRNA inhibits autophagy acting as a ceRNA or directly increasing the activity of mTORC1 by binding to different proteins of the complex. Please review this part.

Reply: Agree. We corrected this part.

  1. This review offers a large amount of information that has to be better defined and organized. The authors have to create well defined subsections (with a title) and present the results whether by diseases or by lncRNAs/ceRNAs (probably the best option as authors did in the 4th section). This will add more clarity to the review. For example, in the second section (“CeRNAs regulated autophagy in cancer”) sometimes paragraphs are dedicated to lncRNAs (p.3, last paragraph), sometimes to groups of proteins (p. 4 second paragraph), sometimes to the disease (p.4 3rd paragraph). In addition, in p.4 last paragraph, authors give an overview of lncRNAs that do not act as ceRNAs. Perhaps this last paragraph could be extended to PVT1 for example that besides its role as a ceRNA also has a direct effect on ULK1 (p.4 first paragraph). The same occurs in section 5, where paragraphs describing lncRNAs in autophagy and metastasis are mixed with others on the role of hypoxia or lncRNAs contained in exosomes. Perhaps these two paragraphs could be placed at the end of the section after describing the different lncRNAs affecting autophagy and metastasis.

Reply: Agree. We have addressed all these points in the text accordingly.

  1. An important role of reviews besides giving an updated vision of the subject is to also give a critical view on the literature together with the most prominent perspectives. The first objective is reached with many very recent data (2020) and this is certainly a very positive point in this review, however I miss a more personal involvement of the authors concerning the discussion of the points addressed in this review.

Reply: Agree. We expanded our personal involvement in this review. Please see last section ‘Conclusion and future perspectives’.

  1. I would add a specific section describing the potential use of ceRNAs or miRNAs as therapeutic targets or tools for autophagy in the context of cancer.

Reply: Agree. We added section 6 ‘ceRNAs as therapeutic targets and biomarkers in cancers’.

Reviewer 2 Report

The work is very well structured, with a logical sequence that describes from the concepts and classification of autophagy, to different types of ncRNAs, focusing on ceRNAs and their mechanism of action through microRNAs.

Figure 1 is very well presented, there are only a couple of details that could be improved. In the text miR-15b-5p is mentioned, but in the figure it appears as miR-15b-3p, and its regulation through MALAT1 on mTOR.

As the role of PVT1 / miR-20a-5p and its regulation on ULK1 are mentioned, but also in the text, miR-485 and miR-365 are mentioned, it would be good to add them to the figure.

Most of the article focuses on describing the relationship between lncRNAs that act on genes related to autophagy, and that an increase in protective autophagy in various types of cancers is related to an increase in metastasis and resistance to chemotherapy. It would be good to highlight if this type of relationship is more typical for some specific types of cancers and what about the expectations for the future.

In point 5 when talking about niche pre-metastasis, then you refers to pro-metastasis nuche in text, clarify the terminology.

Regarding the autophagy regulation pathway from CAF in the figure 3, the text shows a relationship between HOTAIR and miR-454-3p, but miR-93 appears in the figure, is that correct?

The bibliography of the studies is quite up-to-date. It would be good to point out if there are any studies that are aimed at testing the inhibition effect of certain lncRNAs not only in cell cultures but also in live animals and if there is something about a possible clinical study. The topic developed has a great importance for clinical outcomes.

Review bibliography, many abbreviated authors come out, I suppose this will be corrected for the final version.

Author Response

Referee: 2

The work is very well structured, with a logical sequence that describes from the concepts and classification of autophagy, to different types of ncRNAs, focusing on ceRNAs and their mechanism of action through microRNAs.

  1. Figure 1 is very well presented, there are only a couple of details that could be improved. In the text miR-15b-5p is mentioned, but in the figure it appears as miR-15b-3p, and its regulation through MALAT1 on mTOR.

Reply: Agree. We corrected the mistake in figure 1.

  1. As the role of PVT1 / miR-20a-5p and its regulation on ULK1 are mentioned, but also in the text, miR-485 and miR-365 are mentioned, it would be good to add them to the figure

Reply: Agree. PVT1 functioned as ceRNA to regulate ATG3 expression has been added to the figure. PVT1 regulating autophagy through sponging miR-365 was not well-established and therefore, was not added.

  1. Most of the article focuses on describing the relationship between lncRNAs that act on genes related to autophagy, and that an increase in protective autophagy in various types of cancers is related to an increase in metastasis and resistance to chemotherapy. It would be good to highlight if this type of relationship is more typical for some specific types of cancers and what about the expectations for the future.

Reply: Agree. We summarized the roles of ceRNAs in HCC, colorectal cancer and pancreatic cancers in Table 2 ‘CeRNAs regulate autophagy in HCC, colorectal cancer and pancreatic cancer’ and future expectations in Section 6 ‘ceRNAs as therapeutic targets and biomarkers in cancers’.

  1. In point 5 when talking about niche pre-metastasis, then you refers to pro-metastasis nuche in text, clarify the terminology.

Reply: The ‘pro-metastasis’ has been corrected into ‘pre-metastasis’.

  1. Regarding the autophagy regulation pathway from CAF in the figure 3, the text shows a relationship between HOTAIR and miR-454-3p, but miR-93 appears in the figure, is that correct?

Reply: “miR-454-3p”, not “miR-93) is given in the figure.

  1. The bibliography of the studies is quite up-to-date. It would be good to point out if there are any studies that are aimed at testing the inhibition effect of certain lncRNAs not only in cell cultures but also in live animals and if there is something about a possible clinical study. The topic developed has a great importance for clinical outcomes.

Reply: Agree. We summarized the information in Table 2 “CeRNAs regulate autophagy in HCC, colorectal cancer and pancreatic cancer”. The topic has been developed in Section 6 ‘ceRNAs as therapeutic targets and biomarkers in cancers’ and the last section of conclusion and prospective.

  1. Review bibliography, many abbreviated authors come out, I suppose this will be corrected for the final version.

Reply: Agree. The abbreviated authors have been corrected.

Reviewer 3 Report

The manuscript details a thoroughly referenced account of competitive endogenous sponging of microRNAs that have been reported to occur that is of relevance to autophagy in the context of cancer.

Referencing is extensive and the style of english is acceptable (some minor issues to do with correct tense etc on occasion, but nothing I would require correction for).

Although this is a thoroughly well referenced article, I do have several suggestions and one over-riding concern / comment which I would implore the authors to consider:

Minor Points:

1) A figure to do with section 1 would be helpful in illustrating the process of autophagy and the genes involved.

2) LncRNAs and circRNAs are used throughout and the review would benefit from a separate (even if short) section highlighting the different types of RNAs that could serve as ceRNAs (lncRNAs, circRNAs, pseudogene RNAs, mRNAs etc)

Major point:

3) It is my reading of the field that extensive controversy exists regarding the stoichiomentry required for an RNA to truly act as a miRNA sponge at endogenous levels. There are MANY papers describing sponging, but for the most part, these are often reliant on over-expression and fail to adequately deal with the concern that a given "sponge" at endogenous (moderate) levels of expression, often only possessing one or a few target sites for a given microRNA, represents only a small proportion of binding sites in a cells transcriptomic millieu. This is especially the case with circRNAs which are often at very low levels of expression. As such, how much of the effects one can show with over-expression are actually of biological relevance endogenously?

It is my understanding this is a significant issue in the field but this review (and many others like it) completely ignore this area. Reviews by Thomson and Dinger, Nature Reviews Genetics, 2016 and Li, Ma and Li, WIREs RNA, 2019 for example do discuss these questions. 

In my view, a good review will provide some form of commentary regarding these issues as opposed to simply tabulating a list of references. If the authors are able to add this to this manuscript, it will improve it significantly (in my opinion).

Author Response

Referee: 3

The manuscript details a thoroughly referenced account of competitive endogenous sponging of microRNAs that have been reported to occur that is of relevance to autophagy in the context of cancer.

Referencing is extensive and the style of english is acceptable (some minor issues to do with correct tense etc on occasion, but nothing I would require correction for).

Although this is a thoroughly well referenced article, I do have several suggestions and one over-riding concern / comment which I would implore the authors to consider:

Minor Points:

1) A figure to do with section 1 would be helpful in illustrating the process of autophagy and the genes involved.

Reply: Agree.

 2) LncRNAs and circRNAs are used throughout and the review would benefit from a separate (even if short) section highlighting the different types of RNAs that could serve as ceRNAs (lncRNAs, circRNAs, pseudogene RNAs, mRNAs etc)    

Reply: Agree.

Major point:

  • It is my reading of the field that extensive controversy exists regarding the stoichiomentry required for an RNA to truly act as a miRNA sponge at endogenous levels. There are MANY papers describing sponging, but for the most part, these are often reliant on over-expression and fail to adequately deal with the concern that a given "sponge" at endogenous (moderate) levels of expression, often only possessing one or a few target sites for a given microRNA, represents only a small proportion of binding sites in a cells transcriptomic millieu. This is especially the case with circRNAs which are often at very low levels of expression. As such, how much of the effects one can show with over-expression are actually of biological relevance endogenously?

Reply: Agree. These limitations have been discussed in the conclusion and future perspectives section.

4) It is my understanding this is a significant issue in the field but this review (and many others like it) completely ignore this area. Reviews by Thomson and Dinger, Nature Reviews Genetics, 2016 and Li, Ma and Li, WIREs RNA, 2019 for example do discuss these questions.

Reply: Agree. We have discussed these issues in the conclusion and future perspectives section and cited these elaborate references.

5) In my view, a good review will provide some form of commentary regarding these issues as opposed to simply tabulating a list of references. If the authors are able to add this to this manuscript, it will improve it significantly (in my opinion).

Reply: Agree. Please also see section 6 and the section of conclusion and future perspectives.

Round 2

Reviewer 1 Report

The authors have responded to all suggestions.

A few number of typo errors are still present. For example:

line 351: the authors meant "several"?

Line 416: "..involving autophagy are oncogenes and are highly expressed..."

line 425: "oncogenic"

line 428: "...theyand the related patway..", please rephrase

line 436: "...owing the lack of tissue..."

line 485: "stoichiometry"

line 514: "point of view"